

# A survey of secure middleware for the Internet of Things

Paul Fremantle and Philip Scott

School of Computing, University of Portsmouth, Portsmouth, United Kingdom

## ABSTRACT

The rapid growth of small Internet connected devices, known as the Internet of Things (IoT), is creating a new set of challenges to create secure, private infrastructures. This paper reviews the current literature on the challenges and approaches to security and privacy in the Internet of Things, with a strong focus on how these aspects are handled in IoT middleware. We focus on IoT middleware because many systems are built from existing middleware and these inherit the underlying security properties of the middleware framework. The paper is composed of three main sections. Firstly, we propose a matrix of security and privacy threats for IoT. This matrix is used as the basis of a widespread literature review aimed at identifying requirements on IoT platforms and middleware. Secondly, we present a structured literature review of the available middleware and how security is handled in these middleware approaches. We utilise the requirements from the first phase to evaluate. Finally, we draw a set of conclusions and identify further work in this area.

## INTRODUCTION

The Internet of Things (IoT) was originally coined as a phrase by Kevin Ashton in 1990 (*Ashton, 2009*), with reference to "taggable" items that used Radio Frequency Identification Devices (RFID) to become electronically identifiable and therefore amenable to interactions with the Internet. With the ubiquity of cheap processors and System-on-Chip based devices, the definition has expanded to include wireless and Internet-attached sensors and actuators, including smart meters, home automation systems, Internet-attached set-top-boxes, smartphones, connected cars, and other systems that connect the physical world to the Internet either by measuring it or affecting it.

There are a number of definitions of IoT. For the purposes of this work, we will define it in the following way. An IoT *device* is a system that contains either *sensors* or *actuators* or both and supports connection to the Internet either directly or via some intermediary. A sensor is a subcomponent of a device that measures some part of the world, allowing the device to update Internet and Cloud systems with this information. A sensor may be as simple as a button (e.g., Amazon Dash Button), but more complex sensors widely deployed include weather sensors (barometers, anemometers, thermometers), accelerometers and GPS units, light sensors, air quality sensors, people-counters, as well as medical sensors (blood sugar, heart rate, etc.), industrial sensors (production line monitoring, etc.) and many more. Actuators are electronically controlled systems that affect the physical world.

Corresponding author
Paul Fremantle,
paul.fremantle@port.ac.uk

These includes lights, heaters, locks, motors, pumps, relays and so forth. Therefore the IoT is the network of such devices together with the Internet systems that are designed to interoperate and communicate with those devices, including the websites, cloud servers, gateways and so forth.

The number of IoT devices has grown rapidly, with a recent estimate suggesting that there were 12.5 billion Internet attached devices in 2010 and a prediction of 50 billion devices by 2020 (*Evans 2011*). This brings with it multiple security challenges:

- These devices are becoming more central to people's lives, and hence the security is becoming more important.
- Many IoT devices collect Personally Identifiable Information (PII) which may lead to potential privacy concerns.
- Because devices can affect the physical world, there are potential attacks that may have greater impact than purely virtual attacks.
- These devices, due to size and power limitations, may not support the same level of security that we would expect from more traditional Internet connected systems.
- The sheer scale and number of predicted devices will create new challenges and require new approaches to security.

Because of the pervasive and personal nature of IoT, privacy and security are important areas for research. In 2016, more than 100,000 IoT devices were conjoined into a hostile botnet named Mirai that attacked the DNS servers of the east coast of the US (*Wei, 2016*). The total attack bandwidth of this system was measured at more than 600 Gbps. In fact, the number of devices attacked was a small number compared to the potential: previous research (*National Vulnerability Database, 2014*) has identified several million devices that are available for attack.

Therefore there is a strong motivation to find approaches to improve and enhance the security and privacy of the IoT. Many IoT projects use existing *platforms*, also known as *middleware* to build upon. The *Oxford English Dictionary (2017)* defines middleware as:

Software that acts as a bridge between an operating system or database and applications, especially on a network.

Such systems can either improve security or reduce it: if the platform is built with privacy and security in mind then such systems can embed best-practices and enable system designers to rapidly create secure systems. If platforms are built without security, or security is added as an after-thought, then it is possible that not only does the platform encourage the creation of insecure, privacy-negating systems, but also that it may make it more difficult to add security when problems are found. The creation of systems with security and privacy as a key design principle is known as Privacy By Design (PBD) (*Cavoukian, 2008*).

The rest of this work is laid out as follows. In 'Approach and Methodology, we outline the research approach and methodology used for the survey. In 'Matrix Evaluation' we evaluate threats for security and privacy using a matrix model. In 'Three Layer Privacy Model', we use a three-layer model to evaluate IoT privacy. From these models we present

a set of requirements for IoT privacy and security in 'Summary of the Review of Security Issues'. In 'Secure Middleware for the Internet of Things' we outline the structured survey of IoT middleware systems. In 'Secured Systems', we identify 19 secure middleware systems and look at the security and privacy characteristics of each, using the previously identified requirements as a guide. In 'Summary of IoT Middleware Security' we summarise the findings of the survey. Finally, in 'Discussion' we look at the conclusions, contributions and further work in this area.

## APPROACH AND METHODOLOGY

In order to understand the security threats against the Internet of Things, we need to take an approach to classifying threats. The most widely used ontology of security threats is the Confidentiality Integrity Availability (CIA) triad (*Pfleeger & Pfleeger, 2002*) which has been extended over the years. The extended ontology is referred to as the "CIA Plus" (CIA+) model (*Simmonds, Sandilands & Van Ekert, 2004*). In the course of reviewing the available literature and approaches to IoT security, we have created a proposed expansion of the existing framework that we believe works better in the IoT space. In particular, we propose a new ontology based on a matrix of evaluation where we look at each of the classic security challenges in three different aspects: device/hardware, network, and cloud/server-side. In some cells in this matrix, we have not identified any areas where the IoT space presents new challenges: in other words, whilst the domain space covered by these cells contains security challenges, those challenges are no different from existing Web and Internet security challenges in that domain. In those cells we can say that the challenges are "unchanged". In other cells we specifically identify those challenges that are significantly modified by the unique nature of the Internet of Things.

In addition to the matrix, we utilise the *Three Layer Privacy Model* from *Spiekermann & Cranor (2009)* to explore privacy concerns in more detail.

Together, the matrix and three-layer model are then used to inform a set of requirements on IoT middleware. In the second part of this work, we use a structured survey methodology to identify a set of middleware designed to support IoT systems. We start with a specific set of search terms used against a meta-search engine to search across multiple databases. Then we reviewed the abstracts of each identified paper and from these we identified a number of middleware systems. Once the middleware systems we identified, we did not confine ourselves to the identified papers but also reviewed Open Source code, architecture documents and other resources. We evaluate each of the middleware systems against the identified requirements from the matrix evaluation.

The contributions of this paper are:

- A matrix model for evaluating threats to IoT systems.
- A structured literature review of security of middleware systems for IoT.

## MATRIX EVALUATION

Table 1 shows the matrix we will use for evaluating security challenges. In each cell we summarise the main challenges that are *different* in the IoT world or at least exacerbated

**Table 1** Matrix of security challenges for the IoT.

| Security characteristic | A. Device/Hardware | B. Network | C. Cloud/Server-side |
| --- | --- | --- | --- |
| 1. Confidentiality | A1. Hardware attacks | B1. Encryption with low capability devices | C1. Privacy data leaks fingerprinting |
| 2. Integrity | A2. Spoofing; Lack of attestation | B2. Signatures with low capability devices Sybil attacks | C2. No common device identity |
| 3. Availability | A3. Physical attacks; | B3. Unreliable networks, DDoS, radio jamming | C3. DDoS (as usual) |
| 4. Authentication | A4. Lack of UI, default passwords, hardware secret retrieval | B4. Default passwords, lack of secure identities | C4. No common device identity, insecure flows |
| 5. Access Control | A5. Physical access; Lack of local authentication | B5. Lightweight distributed protocols for access control | C5. Inappropriate use of traditional ACLs, device shadow |
| 6. Non-Repudiation | A6. No secure local storage; No attestation, forgery | B6. Lack of signatures with low capability devices | C6. Lack of secure identity and signatures |

by the challenges of IoT compared to existing Internet security challenges. We will explore each cell in the matrix in detail below. Each of the cells is given a designation from *A1* to *C6* and these letters are used as a key to refer to the cells below.

The three aspects (Hardware/Device, Network, Cloud/Server) were chosen because as we read the available literature these areas became clear as a way of segmenting the unique challenges within the context of the IoT. These form a clear logical grouping of the different assets involved in IoT systems. We will provide a quick overview of each area before we look in detail at each cell of the matrix.

**Device and Hardware**

IoT devices have specific challenges that go beyond those of existing Internet clients. These challenges come from: the different form factors of IoT devices, from the power requirements of IoT devices, and from the hardware aspects of IoT devices. The rise of cheap mobile telephony has driven down the costs of 32-bit processors (especially those based around the ARM architecture (*Furber, 1996*)), and this is increasingly creating lower cost microcontrollers and System-on-Chip (SoC) devices based on ARM. However, there are still many IoT devices built on 8-bit processors, and occasionally, 16-bit (*Vieira et al., 2003*). In particular the open source hardware platform Arduino (*Arduino, 2015*) supports both 8-bit and 32-bit controllers, but the 8-bit controllers remain considerably cheaper and at the time of writing are still widely used.

The challenges of low-power hardware mean that certain technologies are more or less suitable. In the details of each cell below we will address specific details as they pertain to security. In addition, there are specific protocols and approaches designed for IoT usage that use less power and are more effective. In *Gligorić, Dejanović & Krčo (2011)* there is a comparison of eXtensible Markup Language (XML) parsing with binary alternatives. The processing time on a constrained device is more than a magnitude slower using XML, and that the heap memory used by XML is more than 10 Kb greater than with binary formats. These improvements result in a 15% saving in power usage in their tests. XML security standards such as XML Encryption and the related WS-Encryption standard have significant problems in an IoT device model. For example, any digital signature

in XML Security needs a process known as XML Canonicalisation (XML C14N). XML Canonicalisation is a costly process in both time and memory. *Binna (2008)* shows that the memory usage is more than 10× the size of the message in memory (and XML messages are already large for IoT devices). We looked for any work on implementing WS-Security on Arduino, ESP8266 or Atmel systems (which are common targets for IoT device implementations) without success. XML performance on small devices can be improved using Efficient XML Interchange (EXI), which reduces network traffic (*Levä, Mazhelis & Suomi, 2014*).

**Network**

IoT devices may use much lower power, lower bandwidth networks than existing Internet systems. Cellular networks often have much higher latency and more "dropouts" than fixed networks (*Chakravorty, Cartwright & Pratt, 2002*). The protocols that are used for the Web are often too data-intensive and power-hungry for IoT devices. Network security approaches such as encryption and digital signatures are difficult and in some cases impractical in small devices. New low-power, low-bandwidth networks such as LoRaWan (https://www.lora-alliance.org/) are gaining significant traction.

There have been some limited studies comparing the power usage of different protocols. In *Levä, Mazhelis & Suomi (2014)* there is comparison of using Constrained Application Protocol (CoAP) with EXI against HyperText Transfer Protocol (HTTP), showing efficiency gains in using CoAP. In *Nicholas (2012)*, MQTT over TLS is shown to use less power than HTTP over TLS in several scenarios. In *Thangavel et al. (2014)*, there is a comparison of network traffic between CoAP and Message Queueing Telemetry Transport (MQTT) showing that each performs better in different scenarios, with similar overall performance. This is an area where more study is clearly needed, but we can draw conclusions that traditional protocols such as Simple Object Access Protocol (SOAP)/HTTP are unsuited to IoT usage.

**Cloud/Server-Side** While many of the existing challenges apply here, there are some aspects that are exacerbated by the IoT for the server-side or cloud infrastructure. These include: the often highly personal nature of data that is being collected and the requirement to manage privacy; the need to provide user-managed controls for access; and the lack of clear identities for devices making it easier to spoof or impersonate devices.

## A1: Device confidentiality

Hardware devices have their own challenges for security. There are systems that can provide tamper-proofing and try to minimise attacks, but if an attacker has direct access to the hardware, they can often break it in many ways. For example, there are devices that will copy the memory from flash memory into another system (known as *NAND Mirroring*). Code that has been secured can often be broken with Scanning Electron Microscopes. Skorobogatov from Cambridge University has written a comprehensive study (*Skorobogatov, 2005*) of many semi-invasive attacks that can be done on hardware. Another common attack is called a *side-channel attack* (*Yan, 2008*; *Lomne et al., 2011*)

where the power usage or other indirect information from the device can be used to steal information. This means that it is very difficult to protect secrets on a device from a committed attacker.

A specific outcome of this is that designers should not rely on obscurity to protect devices. A clear example of this was the Mifare card used as the London Oyster card and for many other authentication and smart-card applications. The designers created their own cryptographic approach and encryption algorithms. Security researchers used a number of techniques to break the obscurity, decode the algorithm, find flaws in it and create a hack that allowed free transport in London as well as breaking the security on a number of military installations and nuclear power plants (*Garcia et al., 2008*). Similarly, relying on the security of a device to protect a key that is used across many devices is a significant error. For example, the encryption keys used in DVD players and XBoX gaming consoles (*Steil, 2005*) were broken meaning that all devices were susceptible to attack.

A related issue to confidentiality of the data on the device is the challenges inherent in updating devices and pushing keys out to devices. The use of Public Key Infrastructure (PKI) requires devices to be updated as certificates expire. The complexity of performing updates on IoT devices is harder, especially in smaller devices where there is no user interface. For example, some devices need to be connected to a laptop in order to perform updates. Others need to be taken to a dealership or vendor. The distribution and maintenance of certificates and public-keys onto embedded devices is complex (*Watro et al., 2004*). In *Park & Kang (2015)* a novel approach to supporting mutual authentication in IoT networks is proposed. However, this model assumes that each device has a secure, shared key (called *kIR*) already deployed and managed into every device. As discussed above, ensuring this key is not compromised is a challenge, as the authors admit: "However, further research is required to realize the secure sharing of *kIR*."

In addition, sensor networks may be connected intermittently to the network resulting in limited or no access to the Certificate Authority (CA). To address this, the use of threshold cryptographic systems that do not depend on a single central CA has been proposed (*Yi & Kravets, 2002*), but this technology is not widely adopted: in any given environment this would require many heterogeneous Things to support the same threshold cryptographic approach. This requires human intervention and validation, and in many cases this is another area where security falls down. For example, many situations exist where security flaws have been fixed but because devices are in homes, or remote locations, or seen as appliances rather than computing devices, updates are not installed (*Hill, 2013*). The Misfortune Cookie (*Point, 2014*) demonstrates that even when security fixes are available, some manufacturers do not make them available to customers and continue to ship insecure systems. It is clear from the number of publicised attacks (*McDaniel & McLaughlin, 2009*; *Khurana et al., 2010*; *Hill, 2013*) that many device designers have not adjusted to the challenges of designing devices that will be connected either directly or indirectly to the Internet.

A further security challenge for confidentiality and hardware is the fingerprinting of sensors or data from sensors. In *Bojinov et al. (2014)* it has been shown that microphones, accelerometers and other sensors within devices have unique "fingerprints" that can

uniquely identify devices. Effectively there are small random differences in the physical devices that appear during manufacturing that can be identified and used to recognise individual devices across multiple interactions.

## B1: Network confidentiality

The confidentiality of data on the network is usually protected by encryption of the data. There are a number of challenges with using encryption in small devices. Performing public key encryption on 8-bit microcontrollers has been enhanced by the use of Elliptic Curve Cryptography (ECC) (*Koblitz, 1987*; *Miller, 1986*). ECC reduces the time and power requirements for the same level of encryption as an equivalent Rivest, Shamir, and Adleman public key cryptography (RSA) public-key encryption (*Rivest, Shamir & Adleman, 1978*) by an order of magnitude (*Gura et al., 2004*; *Sethi, 2012*; *Sethi, Arkko & Keranen, 2012*): RSA encryption on constrained 8-bit microcontrollers may take minutes to complete, whereas similar ECC-based cryptography completes in seconds. However, despite the fact that ECC enables 8-bit microcontrollers to participate in public-key encryption systems, in many cases it is not used. We can speculate as to why this is: firstly, as evidenced by *Sethi, Arkko & Keranen (2012)*, the encryption algorithms consume a large proportion of the available ROM on small controllers. Secondly, there is a lack of standard open source software. For example, a search that we carried out (on the 21st April 2015) of the popular open source site Github for the words "Arduino" and "Encryption" revealed 10 repositories compared to "Arduino" and "HTTP" which revealed 467 repositories. These 10 repositories were not limited to network level encryption. However, recently an open source library for AES on Arduino (*Landman, 2015*) has made the it more effective to use cryptography on Atmel-based hardware.[1]

While ECC is making it possible for low-power devices to be more efficient in performing cryptography operations, in 2015 the NSA made an unprecedented warning against ECC (https://threatpost.com/nsas-divorce-from-ecc-causing-crypto-hand-wringing/115150/). We don't yet know why, as of the time of writing. There are differing theories. One known issue with both Prime Numbers and Elliptic Curves is Quantum Computing. In Quantum computers, instead of each bit being 0 or 1, each *qubit* allows a superposition of both 0 and 1, allowing Quantum computers to solve problems that are very slow for classical computers in a fraction of the time. At the moment general purpose Quantum computers are very simple and confined to laboratories, but they are increasing in power and reliability. In 1994, Peter Shor identified an algorithm for Quantum Computers (*Shor, 1999*) that performs prime factorization in polynomial time, which effectively means that most existing Public Key Cryptography (PKC) will be broken once sufficiently powerful Quantum computers come online. Given that most Quantum Computers are as yet ineffective, there is some concern that maybe the problem with ECC is actually based on classical computing, but this is all speculation. One thing that we do know is that ECC is much easier to do on IoT devices, and especially on low-power, 8- or 16-bit systems. Therefore this warning is worrying for IoT developers.

Another key challenge in confidentiality is the complexity of the most commonly used encryption protocols. The standard Transport Layer Security (TLS) (*Dierks, 2008*) protocol

[1] The same search was repeated on the 10th Feb 2017. The number of repositories for "Arduino" and "Encryption" had grown to 21, while for "Arduino" and "HTTP" had reached 941, demonstrating that support for encryption is growing slowly.

can be configured to use ECC, but even in this case the handshake process requires a number of message flows and is sub-optimal for small devices as documented in *Koschuch, Hudler & Krüger (2010)*. *Perelman & Ersue (2012)* has argued that using TLS with Pre Shared Key (PSK) improves the handshake. PSK effectively allows TLS to use traditional symmetric cryptography instead of Public Key (assymetric) cryptography. However, they fail to discuss in any detail the significant challenges with using PSK with IoT devices: the fact that either individual symmetric keys need to be deployed onto each device during the device manufacturing process, or the same key re-used. In this case there is a serious security risk that a single device will be broken and thus the key will be available.

Some IoT devices use User Datagram Protocol (UDP) instead of the more commonly used Transport Control Protocol (TCP). Both protocols are supported on the Internet. UDP is unreliable, and is typically better suited to local communications on trusted networks. It is more commonly used between IoT devices and gateways rather than directly over the Internet, although, like all generalisations there are exceptions to this rule. TLS only works with TCP, and there is an alternative protocol for UDP. Datagram Transport Layer Security (DTLS) (*Rescorla & Modadugu, 2006*) provides a mapping of TLS to UDP networks, by adding retransmission and sequencing which are assumed by TLS. While the combination of DTLS and UDP is lighter-weight than TLS and TCP, there is still a reasonably large RAM and ROM size required for this (*Keoh, Kumar & Garcia-Morchon, 2013*), and this requires that messages be sent over UDP which has significant issues with firewalls and home routers, making it a less effective protocol for IoT applications (*Audet & Jennings, 2007*). There is ongoing work at the IETF to produce an effective profile of both TLS and DTLS for the IoT (*Tschofenig & Fossati, 2016*).

A significant area of challenge for network confidentiality in IoT is the emergence of new radio protocols for networking. Previously there were equivalent challenges with Wifi networks as protocols such as Wired Equivalency Privacy (WEP) were broken (*Cam-Winget et al., 2003*), and there are new attacks on protocols such as Bluetooth Low Energy (BLE) (also known as Bluetooth 4.0). For example, while BLE utilises Advanced Encryption Standard (AES) encryption which has a known security profile, a new key exchange protocol was created, which turns out to be flawed, allowing any attacker present during key exchange to intercept all future communications (*Ryan, 2013*). One significant challenge for IoT is the length of time it takes for vulnerabilities to be addressed when hardware assets are involved. While the BLE key exchange issues are addressed in the latest revision of BLE, we can expect it to take a very long time for the devices that encode the flawed version in hardware to be replaced, due to the very large number of devices and the lack of updates for many devices. By analogy, many years after the WEP issues were uncovered, in 2011 a study showed that 25% of wifi networks were still at risk (*Botezatu, 2011*).

Even without concerning the confidentiality of the data, there is one further confidentiality issue around IoT devices in the network and that is confidentiality of the metadata. Many IoT systems rely on radio transmission and in many cases they can be fingerprinted or identified by the radio signature. For example, Bluetooth and Wifi systems use unique identifiers called MAC address (Media Access Control). These can be identified by scanning, and there have been a number of systems deployed to do that,

including in airports and in cities (*Vincent, 2013*). These systems effectively can follow users geographically around. If the user then connects to a system, that fingerprint can be associated with the user and the previously collected location information can be correlated with that user. In a similar attack, security researchers recently found (*Schneier, 2008*) that they could fingerprint cars based on transmissions from tyre pressure monitors, and in addition that they could drive behind a car and from up to 40 feet away they could signal to the driver that the tyre pressure was dangerously low when in fact it wasn't. Such an attack could easily be used to get a driver to stop and leave their car.

In (*Radomirovic, 2010*) a theoretical model of traceability of IoT devices and particularly Radio Frequency Identification Device (RFID) systems is proposed in order to prevent unauthorised data being accessible. A protocol that preserves the concept of untraceability is proposed.

Many of the same references and issues apply to section *B2* where we look at the use of digital signatures with low power devices.

## C1: Cloud confidentiality

In the main, the issues around Cloud confidentiality are the same as the issues in non-IoT systems. There are however, some key concerns over privacy that are unique to the Internet of Things. For example, the company Fitbit (*Fitbit, 2015*) made data about users sexual activity available and easily searchable online (*Zee, 2011*) by default. There are social and policy issues regarding the ownership of data created by IoT devices (*Rendle, 2014*; *Murphy, 2014*). We address these issues in more detail in cell *C5* where we look at the access control of IoT data and systems in the cloud and on the server-side.

A second concern that is exacerbated by the Internet of Things are concerns with correlation of data and metadata, especially around *de-anonymisation*. In *Narayanan & Shmatikov (2008)* it was shown that anonymous metadata could be de-anonymized by correlating it with other publicly available social metadata. This is a significant concern with IoT data. This is also closely related to the fingerprinting of sensors within devices as discussed in cell *A1*. An important model for addressing these issues in the cloud are systems that filter, summarise and use *stream-processing* technologies to the data coming from IoT devices before this data is more widely published. For example, if we only publish a summarised co-ordinate rather than the raw accelerometer data we can potentially avoid fingerprinting de-anonymisation attacks.

In addition, an important concern has been raised in the recent past with the details of the government sponsored attacks from the US National Security Agency (NSA) and UK Government Communications Headquarters (GCHQ) that have been revealed by Edward Snowden (*Card, 2015*). These bring up three specific concerns on IoT privacy and confidentiality.

The first concern is the revelations that many of the encryption and security systems have had deliberate backdoor attacks added to them so as to make them less secure (*Larson, Perlroth & Shane, 2013*). The second concern is the revelation that many providers of cloud hosting systems have been forced to hand over encryption keys to the security

services (*Levinson, 2014*). The third major concern is the revelations on the extent to which metadata is utilised by the security services to build up a detailed picture of individual users (*Ball, 2013*).

The implications of these three concerns when considered in the light of the Internet of Things is clear: a significantly deeper and larger amount of data and metadata will be available to security services and to other attackers who can utilize the same weaknesses that the security services compromise.

## A2: Integrity & hardware/device

The concept of integrity refers to maintaining the accuracy and consistency of data. In this cell of the matrix, the challenges are in maintaining the device's code and stored data so that it can be trusted over the lifecycle of that device. In particular the integrity of the code is vital if we are to trust the data that comes from the device or the data that is sent to the device. The challenges here are viruses, firmware attacks and specific manipulation of hardware. For example, *Goodin (2013)* describes a worm attack on router and IoT firmware, where each compromised system then compromises further systems, leaving behind a slew of untrustworthy systems.

The traditional solution to such problems is attestation (*Sadeghi & Stüble, 2004*; *Brickell, Camenisch & Chen, 2004*; *Seshadri et al., 2004*). Attestation is important in two ways. Firstly, attestation can be used by a remote system to ensure that the firmware is unmodified and therefore the data coming from the device is accurate. Secondly, attestation is used in conjunction with hardware-based secure storage (Hardware Security Managers, as described in *Deitel (1984)*) to ensure that authentication keys are not misused. The model is as follows.

In order to preserve the security of authentication keys in a machine where human interaction is involved, the user is required to authenticate. Often the keys are themselves encrypted using the human's password or a derivative of the identification parameters. However, in an unattended system, there is no human interaction. Therefore the authentication keys need to be protected in some other way. Encryption on its own is no help, because the encryption key is then needed and this becomes a circular problem. The solution to this is to store the authentication key in a dedicated hardware storage. However, if the firmware of the device is modified, then the modified firmware can read the authentication key, and offer it to a hacker or misuse it directly. The solution to this is for an attestation process to validate the firmware is unmodified before allowing the keys to be used. Then the keys must also be encrypted before sending them over any network.

These attestation models are promoted by groups such the Trusted Computing Group (*TCG, 2015*), and Samsung Knox (*Samsung, 2015*). These rely on specialized hardware chips such as the Atmel AT97SC3204 (*Atmel, 2015*) which implement the concept of a Trusted Platform Module (TPM) (*Morris, 2011*). There is research into running these for Smart Grid devices (*Paverd & Martin, 2012*). However, whilst there is considerable discussion of using these techniques with IoT, during our literature review we could not find evidence of any real-world devices apart from those based on mobile-phone platforms (e.g., phones and tablets) that implemented trusted computing and attestation.

## B2: Network integrity

Maintaining integrity over a network is managed as part of the public-key encryption models by the use of digital signatures. The challenges for IoT are exactly those we already identified in cell *B1* above where we described the challenges of using encryption from low-power IoT devices.

However, there is a further concern with IoT known as the Sybil Attack (*Douceur, 2002*). A Sybil attack[2] is where a peer-to-peer network is taken over when an attacker creates a sufficiently large number of fake identities to persuade the real systems of false data. A Sybil attack may be carried out by introducing new IoT devices into a locality or by suborning existing devices. For example, it is expected that autonomous cars may need to form local ephemeral peer-to-peer networks based on the geography of the road system. A significant threat could be provided if a Sybil attack provided those cars with incorrect data about traffic flows.

## C2: Cloud integrity

The biggest concern in this area is the lack of common concepts and approaches for device identity. Integrity relies on identity—without knowing who or what created data, we cannot trust that data. We address this in *A4, B4* and *C4*. One specific aspect of trust in cloud for IoT scenarios is where the device lacks the power to participate in trust and must therefore trust the cloud server. One key example of this is where a *blockchain* (*Nakamoto, 2012*) is being used in respect of IoT devices. Blockchains are cryptographically secure ledgers that typically require a significant amount of memory, disk space and processor power to work (*Bitcoin, 2017*). These requirements go beyond typical IoT devices and even beyond more powerful systems in IoT networks such as hubs. One option to address this is to use remote attestation, but as yet there is little or no work in this space.

## A3: Hardware availability

One of the significant models used by attackers is to challenge the availability of a system, usually through a Denial of Service (Dos) or Distributed Denial of Service (DDos) attack. DoS attacks and availability attacks are used in several ways by attackers. Firstly, there may be some pure malicious or destructive urge (e.g., revenge, commercial harm, share price manipulation) in bringing down a system. Secondly, availability attacks are often used as a pre-cursor to an authentication or spoofing attack.

IoT devices have some different attack vectors for availability attacks. These include resource consumption attacks (overloading restricted devices), physical attacks on devices. A simple availability attack on an IoT device might be to force it to use more power (e.g., by initiating multiple key exchanges over Bluetooth) and thereby draining the battery. Another even more obvious availability challenge would be to simply physically destroy a device if it is left in a public or unprotected area.

## B3: Network availability

There are clearly many aspects of this that are the same as existing network challenges. However, there are some issues that particularly affect IoT. In particular, there are a number of attacks on local radio networks that are possible. Many IoT devices use radio networking

[2]Named after a character in a book who exhibits multiple personality disorder.

(Bluetooth, Wifi, 3G, General Packet Radio Service (GPRS), LoRa and others) and these can be susceptible to radio jamming. In *Mpitziopoulos et al. (2009)* there is a survey of jamming attacks and countermeasures in Wireless Sensor Network (WSN). Another clear area of attack is simply physical access. For example, even wired networks are much more susceptible to physical attacks when the devices are spread widely over large areas.

### C3: Cloud availability

The challenges here are not new. Elsewhere we looked at DoS attacks and DDoS attacks. The biggest challenge here is the use of IoT devices themselves to create the DDoS attack on the server, as in the Mirai botnet.

### A4: Device authentication

We will consider the authentication of the device to the rest of the world in cells *B5* and *C5*. In this cell of the matrix we must consider the challenges of how users or other devices can securely authenticate to the device itself. These are however related: a user may bypass or fake the authentication to the device and thereby cause the device to incorrectly identify itself over the network to other parts of the Internet.

Some attacks are very simple: many devices come with default passwords which are never changed by owners. In a well-publicised example (*Hill, 2013*), a security researcher gained access to full controls of a number of "smart homes". As discussed above, the Mirai attack took control of devices that used default or easily guessed passwords.

Similarly many home routers are at risk through insecure authentication (*Andersson & Szewczyk, 2011*). Such vulnerabilities can then spread to other devices on the same network as attackers take control of the local area network.

A key issue here is the initial *registration* of the device. A major issue with hardware is when the same credential, key, or password is stored on many devices. Devices are susceptible to hardware attacks (as discussed above) and the result is that the loss of a single device may compromise many or all devices. In order to prevent this, devices must either be pre-programmed with unique identifiers and credentials at manufacturing time, or must go through a registration process at setup time. In both cases this adds complexity and expense, and may compromise usability. In *Fremantle, Kopecký & Aziz (2015)* there is a proposal for the use of the OAuth2 Dynamic Client Registration (*Sakimura, Bradley & Jones, 2015*) process to create unique keys/credentials for each device. In *Fremantle & Aziz (2016)* there is a well-defined and secure process for device and user registration that allows users to take control of devices in scenarios where the device itself offers no User Interface (UI) or a very basic UI.

### B4: Network authentication

Unlike browsers or laptops where a human has the opportunity to provide authentication information such as a userid and password, IoT devices normally run unattended and need to be able to power-cycle and reboot without human interaction. This means that any identifier for the device needs to be stored in the program memory (usually SRAM), ROM or storage of the device. This brings two distinct challenges:

- The device may validly authenticate, but the program code may have been changed, and therefore it may behave incorrectly.
- Another device may steal the authentication identifier and may *spoof* the device.

In the Sybil attack (*Newsome et al., 2004*) a single node or nodes may impersonate a large number of different nodes thereby taking over a whole network of sensors. In all cases, attestation is a key defence against these attacks.

Another defence is the use of *reputation* and reputational models to associate a trust value to devices on the network. Reputation is a general concept widely used in all aspects of knowledge ranging from humanities, arts and social sciences to digital sciences. In computing systems, reputation is considered as a *measure* of how trustworthy a system is. There are two approaches to trust in computer networks: the first involves a "black and white" approach based on security certificates, policies, etc. For example, SPINS (*Perrig et al., 2002*), develops a trusted network. The second approach is probabilistic in nature, where trust is based on reputation, which is defined as a probability that an agent is trustworthy. In fact, reputation is often seen as one measure by which trust or distrust can be built based on good or bad past experiences and observations (direct trust) (*Jøsang, Ismail & Boyd, 2007*) or based on collected referral information (indirect trust) (*Abdul-Rahman & Hailes, 2000*).

In recent years, the concept of reputation has shown itself to be useful in many areas of research in computer science, particularly in the context of distributed and collaborative systems, where interesting issues of trust and security manifest themselves. Therefore, one encounters several definitions, models and systems of reputation in distributed computing research (e.g., *Fullam & Barber, 2006*; *Jøsang, Ismail & Boyd, 2007*; *Silaghi, Arenas & Silva, 2007*).

There is considerable work into reputation and trust for wireless sensor networks, much of which is directly relevant to IoT trust and reputation. The Hermes and E-Hermes (*Zouridaki et al., 2007*; *Zouridaki et al., 2009*) systems utilise Bayesian statistical methods to calculate reputation based on how effectively nodes in a mesh network propogate messages including the reputation messages. Similarly, *Chen et al. (2011)* evaluates reputation based on the packet-forwarding trustworthiness of nodes, in this case using fuzzy logic to provide the evaluation framework. Another similar work is *Michiardi & Molva (2002)* which again looks at the packet forwarding reputation of nodes. In IoT, *Aziz et al. (2016)* utilizes the concept of a *Utility Function* to create a reputational model for IoT systems using the MQTT protocol.

## C4: Cloud authentication

The IETF has published a draft guidance on security considerations for IoT (*O. Garcia-Morchon, 2013*). This draft does discuss both the bootstrapping of identity and the issues of privacy-aware identification. One key aspect is that of bootstrapping a secure conversation between the IoT device and other systems, which includes the challenge of setting-up an encrypted and/or authenticated channel such as those using TLS, Host Identity Protocol (HIP) or Diet HIP. HIP (*Moskowitz, 2012b*) is a protocol designed to provide a cryptographically secured endpoint to replace the use of IP addresses, which solves a significant problem—IP-address spoofing—in the Internet. Diet HIP (*Moskowitz, 2012a*)

is a lighter-weight rendition of the same model designed specifically for IoT and Machine to Machine (M2M) interactions. While HIP and Diet HIP solve difficult problems, they have significant disadvantages to adoption. Secure device identity models that work at higher levels in the network stack, such as token-based approaches, can sit side by side with existing IP-based protocols and require no changes at lower levels of the stack. By contrast, HIP and Diet HIP require low-level changes within the IP stack to implement. As they replace traditional IP addressing they require many systems to change before a new device using HIP can successfully work. In addition, neither HIP nor Diet HIP address the issues of federated authorization and delegation.

In *Fremantle (2013)* and *Fremantle et al. (2014)* it is proposed to use *federated* identity protocols such as OAuth2 (*Hammer-Lahav & Hardt, 2011*) with IoT devices, especially around the MQTT protocol (*Locke, 2010*). The IOT-OAS (*Cirani et al., 2015*) work similarly addresses the use of OAuth2 with CoAP. Other related works include the work of Augusto et al. (*Augusto & Correia, 2011*) have built a secure mobile digital wallet by using OAuth together with the XMPP protocol (*Saint-Andre, 2011*). In *Fremantle, Kopecký & Aziz (2015)*, the usage of OAuth2 for IoT devices is extended to include the use of Dynamic Client Registration (*Sakimura, Bradley & Jones, 2013*) which allows each device to have its own unique identity, which we discussed as an important point in the section about cell *A1*.

A contradictory aspect of IoT Authentication is the proposal to use secure *Pseudonyms*. A pseudonym is also sometimes referred to as an *Anonymous Identity*. Effectively, a secure pseudonym is a way of a user securely interacting with a system without giving away their real identity. This overlaps with cell *C5* where we look at access control for cloud systems. We have seen from well-publicised cases that systems may be compromised and offer personal information, even years after that information was originally stored. In one case, two suicides have been attributed to an attack that compromised personal identities (*Baraniuk, 2015*). Pseudonyms are an approach that can be considered to treat the sharing of meta-data as important as sharing of data. Also see 'Three Layer Privacy Model' where we look at another model of privacy.

In *Rotondi, Seccia & Piccione (2011)* a capability-based access system is described that allows anonymous identities to be used. *Bernabe et al. (2014)* provides an Architecture Reference Model for an approach that supports anonymous identities. Neither of these systems separate the provision of anonymous identities from the data-sharing middleware. A concept called Zooko's Triangle (*O'Hearn, 2001*) proposed that it is only possible to support two out of the following three capabilities in a system: human-readable names; decentralised infrastructure; and security. Recent papers, such as *Ali et al. (2016)*, claim that the blockchain construct proves Zooko's hypothesis wrong. In *Hardjono, Smith & Pentland (2014)* the concept of anonymous identities for blockchains is explored, which will have significant impact as blockchains become more prevalent in IoT.

## A5: Device access control

There are two challenges to access control at the device level. Firstly, devices are often physically distributed and so an attacker is likely to be able to gain physical access to

the device. The challenges here (hardware attacks, NAND mirroring, etc.) were already discussed in cell *A1*.

However, there is a further challenge: access control almost always requires a concept of identity. We cannot restrict or allow access except in the most basic ways without some form of authentication to the device. As discussed in our review of cell *A4*, this is a significant challenge. To give a real life example, certain mobile phones have recently started encrypting data based on the user's own lock-screen Personal Identification Number (PIN) code (*Schneier, 2014*). This guarantees the data cannot be read without the user's PIN code. However, using NAND Mirroring, it has been demonstrated that the controls that stop repeated attempts at PIN codes can be overcome (*Skorobogatov, 2016*), with the result that a 4 digit PIN can easily be broken within a reasonable amount of time.

Systems such as Webinos (*Desruelle et al., 2012*) have proposed using policy-based access control mechanisms such as XML Access Control Markup Language (XACML) (*Godik et al., 2002*) for IoT devices. However, XACML is relatively heavyweight and expensive to implement (*Turkmen & Crispo, 2008*), especially in the context of low power devices. To address this, Webinos has developed an engine which can calculate the subset of the policy that is relevant to a particular device. Despite this innovation, the storage, transmission and processing costs of XACML are still high for an IoT device. Another approach based around a standard called UMA is covered in cell *C5*.

## B5: Network access control

There are a number of researchers looking at how to create new lightweight protocols for access control in IoT scenarios. *Mahalle et al. (2012)* describe a new protocol for IoT authentication and access control is proposed based on ECC with a lightweight handshake mechanism to provide an effective approach for IoT, especially in mobility cases. *Hernández-Ramos et al. (2013)* propose a non-centralised approach for access control that uses ECC once again and supports capability tokens in the CoAP protocol.

## C5: Cloud access control

The biggest challenge for privacy is ensuring access control at the server or cloud environment of data collected from the IoT. There is some significant overlap with the area of confidentiality of data in the cloud as well (cell *C1*).

It is argued in *Fremantle et al. (2014)* that existing hierarchical models of access control are not appropriate for the scale and scope of the IoT. There are two main approaches to address this. The first is *policy-based* security models where roles and groups are replaces by more generic policies that capture real-world requirements such as "A doctor may view a patient's record if they are treating that patient in the emergency room". The second approach to support the scale of IoT is user-directed security controls, otherwise known as consent. This is the approach we take in this thesis. In *Tschofenig et al. (2015)* a strong case is made for ensuring that users can control access to their own resources and to the data produced by the IoT that relates to those users. The User Managed Access (UMA) from the Kantara Initiative enhances the OAuth specification to provide a rich environment for users to select their own data sharing preferences (*Kantara Initiative, 2013*). We would argue

strongly that this overall concept of user-directed access control to IoT data is one of the most important approaches to ensuring privacy. In *Tschofenig et al. (2015)*, an approach for using UMA together with OAuth2 is proposed for constrained devices. This approach also addresses cell *A5*. While this approach has a lot of capabilities and power, there is a slow uptake of UMA in real-world services and even less in IoT. We propose that the complexity of this approach is the inhibitor to widespread adoption.

*Winter (2012)* argues that contextual approaches must be taken to ensure privacy with the IoT. Many modern security systems use context and reputation to establish trust and to prevent data leaks. Context-based security (*Montanari, Toninelli & Bradshaw, 2005*) defines this approach which is now implemented by major Web systems including Google and Facebook. This is closely related to reputation-based models which we discussed above.

### A6: Device non-repudiation

The biggest challenge in the non-repudiation network with IoT devices is the challenge of using *attestation* for small devices. Attestation is discussed in detail in cell *A2*. Without attestation, we cannot trust that the device system has not been modified and therefore it is not possible to trust any non-repudiation data from the device.

### B6: Network non-repudiation

The same challenges apply here as discussed in cells *B2* and, *B3*. Non-repudiation on the wire requires cryptography techniques and these are often hindered by resource restrictions on small devices. In *Park, Seok & Park (2007)* a non-repudiation protocol for restricted devices is proposed.

### C6: Cloud non-repudiation

This area is unchanged by the IoT, so we do not discuss it any further.

In the previous eighteen sections we have outlined a significant number of threats and challenges, and used this matrix to assess the most relevant current work in each space. Before summarising this work, we look at an orthogonal model more closely focussed on user privacy. This model will be used in later sections to assess the outcomes of this thesis.

## THREE LAYER PRIVACY MODEL

One area that crosses most or all of the cells in our matrix is the need for a holistic and studied approach to enabling privacy in the IoT. As discussed in a number of cells, there are significant challenges to privacy with the increased data and metadata that is being made available by IoT-connected devices. An approach that has been proposed to address this is *Privacy by Design* (*Cavoukian, 2008*). This approach suggests that systems should be designed from the ground up with the concept of privacy built into the heart of each system. Many systems have added security or privacy controls as ''add-ons'', with the result that unforeseen attacks can occur.

*Spiekermann & Cranor (2009)* offer a model for looking at user privacy. In their model, they identify three spheres: the *User Sphere*, the *Joint Sphere* and the *Recipient Sphere*. The User Sphere is completely in the control of the user (e.g., a laptop). The Joint Sphere refers

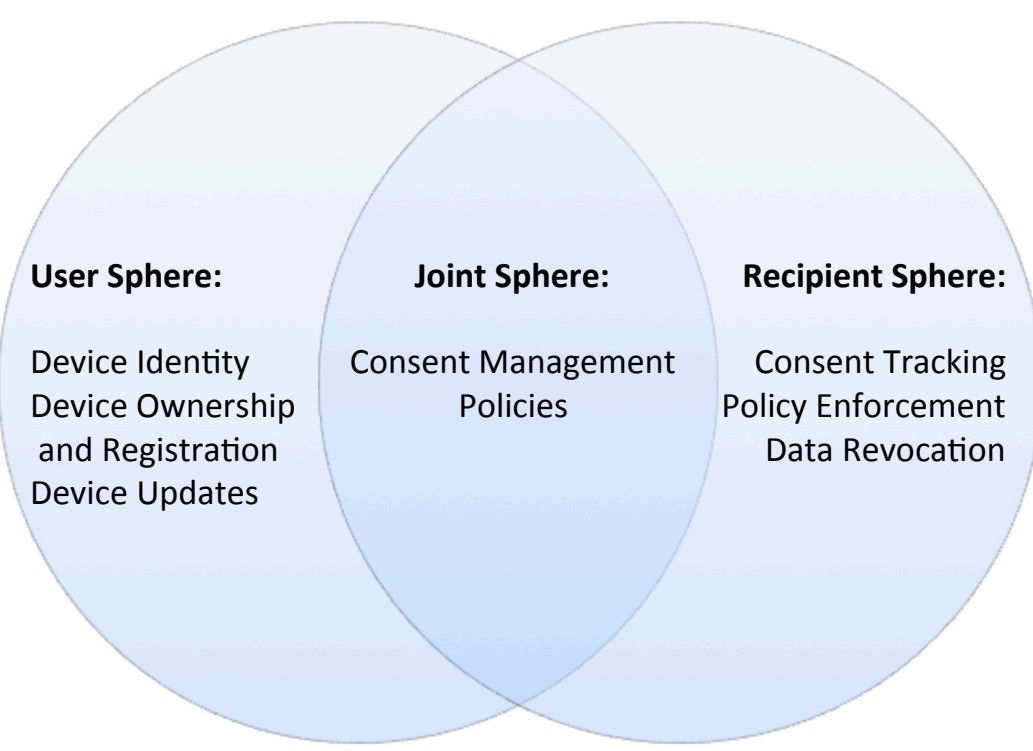

**Figure 1** Three Layer Privacy model applied to IoT.

to areas that may seem to be in the user's control, but may have some significant control by a third-party. For example, a cloud email account may seem like the user can delete emails, but the cloud provider may in fact back these up and keep a copy. Finally, once data has been transferred to a third-party, it is assumed to be in the Recipient Sphere, where the only controls are legal and contractual.

In the model, a device that offers the user full control is firmly in the *User Sphere*. However, we would argue that many current devices are actually in the *Joint Sphere*. This is where the device appears to be in the control of the user but in fact is in the control of a third-party. To give an example, the Google Nest device offers users the opportunity to apply smart heating controls to their house. While a number of user-centred controls give the user the impression that it is in the User Sphere, there are two key reasons to counter this: firstly, the data logged by the device is extensive and cannot be controlled by the user; secondly, the device auto-updates itself based on commands from Google rather than based on user input (*Nest, 2017*).

Using this model, we can propose clear approaches that strengthen each of the privacy and security controls available in each sphere. Figure 1 provides an overview of this model and its applicability to the IoT domain.

## User sphere

Moving privacy and security controls back to the users inherently strengthens the User Sphere and provides greater choice, thereby allowing more secure approaches to flourish.

As discussed above, devices need to have secure identities, and currently these are either not provided, or provided by the device manufacturer.

A second, related issue, is the ownership of devices. The Mirai botnet spread because dictionary attacks allowed attackers to take ownership of devices. Some systems offer models of taking ownership securely (e.g., Bluetooth, Near Field Communication (NFC)). In *Fremantle & Aziz (2016)*, there is a system described where a QR code is used in conjunction with a Web-based system.

A third issue within the User Sphere is updating the device firmware. A number of attacks have originated in lack of updates. One issue is that device manufacturers are incentivised to create new products but not to update old products. In *Tindall (2015)*, a model is proposed whereby devices can pay for updates using a blockchain-based cryptocurrency such as Bitcoin. In the *Fremantle & Aziz (2016)* there is an approach where IoT devices are updated based on the secure identity and consent models used in OAuth2.

## Joint sphere

Recall that the Joint Sphere is the parts of the system where the user has some form of control over their data and systems, but the provider also shares control. For example, a health-monitoring device may upload data to an Internet-based system and then may offer users controls on how they share data. A major change in legislation around this is the European Union's General Data Protection Regulation (GDPR) (*European Commission, 2016*) which requires much stronger consent controls. Many systems offer forms of user consent for sharing data with third parties, but these lack significant requirements. For example, many users are not aware of how to revoke consent. Similarly, there is no clear place a user can identify all the consents they have approved across different devices. Consent is not just about privacy. IoT devices often include actuators that can act based on *Commands*, and the security of a device includes ensuring that only authorised systems can issue commands to devices. We looked at consent.

A related area is that of policies. In this meaning a policy is a computer-readable expression of rights and obligations. For example, a consent approval may refer to a policy: the user might approve sharing of data to a website based on the fact that the website promises not to share the data to any other body. Languages such as XACML (*Godik et al., 2002*) allow complex access control policies to be encoded in XML or JSON. We discussed this in cell *C5*.

## Recipient sphere

The Recipient Sphere is the area where the user's data is now out of their control. Ultimately, the user must rely on legislation or legal contracts in order to maintain control of this data. Of course, it is hard to police this recipient sphere: it is possible that the third-party website will share data following policies. In addition, many organisations have such complex and poorly worded policies that users are unaware of the rights they are giving up to their data. Spotting illicit data shares can possibly be done using a concept of a *Trap Street*. This is the habit that map-makers have of including incorrect data to see if others copy it. Similarly, IoT devices could deliberately share incorrect data to specific parties to see if it leaks out against the agreed policy.

## SUMMARY OF THE REVIEW OF SECURITY ISSUES

In this section we have proposed a widened ontology for evaluating the security issues surrounding the Internet of Things, and examined the existing literature and research in each of the cells of the expanded matrix. We have also related these issues to Spiekermann and Cranor's Three Layer Privacy Model. This is an important basis for the next section where we examine the provisions around security and privacy that are available in available middleware for the Internet of Things.

In reviewing these areas, we identified a list of security properties and capabilities that are important for the security and privacy of IoT. We will use this list in the second part of this paper as columns in a new table where we evaluate a set of middleware on their provision of these capabilities.

**REQ1—integrity and confidentiality.** The requirement to provide integrity and confidentiality is an important aspect in any network and as discussed in cells *A1–B2* there are a number of challenges in this space for IoT.

**REQ2—access control.** Maintaining access control to data that is personal or can be used to extract personal data is a key aspect of privacy. In addition, it is of prime importance with actuators that we do not allow unauthorised access to control aspects of our world.

**REQ2.1—consent.** As described in cells *A5–C5*, traditional hierarchical models of access control are ineffective for personal data and IoT systems. Consent approaches—such as OAuth2 and UMA –are a key requirement.

**REQ2.2—policy-based access control.** As discussed in cells *A5–C5*, policy-based access control models such as XACML enable privacy considerations and rules to be implemented effectively in IoT scenarios, although in many cases models such as XACML are too heavyweight to deploy into devices.

**REQ3—authentication.** As discussed in numerous of the cells, IoT systems need a concept of authentication in order to enable integrity, confidentiality, and access control amongst other requirements.

**REQ3.1—federated identity.** As argued in cells *A4–C4*, there is a clear motivation for the use of federated models of identity for authentication in IoT networks.

**REQ3.2—secure device identity.** Managing the security of devices requires unique credentials to be embedded into each device and secure registration processes as discussed in cell *A4*.

**REQ3.3—anonymous identities.** In order to guard against de-anonymisation and other leakages of personally identifiable information, anonymous identities/pseudonyms can offer individuals clearer consent as to when they wish to actively share their identity, as discussed in *A4*.

**REQ4—attestation.** Attestation is an important technique to prevent tampering with physical devices (as discussed in the cells in column A) and hence issues with integrity of data as well as confidentiality in IoT.

**REQ5—summarisation and filtering.** The need to prevent de-anonymisation is a clear driver for systems to provide summarisation and filtering technologies such as stream processing.

**REQ6—context-based security and reputation.** Many modern security models adapt the security based on a number of factors, including location, time of day, previous history of systems, and other aspects known as context. Another related model is that of the reputation of systems, whereby systems that have unusual or less-than-ideal behaviour can be trusted less using probabilistic models. In both cases there are clear application to IoT privacy as discussed above.

While we consider PBD an important aspect, we argue that it is a *meta-requirement*: it effectively covers the need to implement the major security and privacy requirements from the initial design outwards.

There are of course many other aspects to IoT security and privacy as we have demonstrated in the matrix table and accompanying description of each cell. However, these specific aspects form an effective set of criteria by which to analyse different systems, as we show below in the next section.

## SECURE MIDDLEWARE FOR THE INTERNET OF THINGS

Middleware has been defined as computer software that has an intermediary function between the various applications of a computer and its operating system (*Hanks, 1986*). In our case, we are interested in middleware that is specifically designed or adapted to provide capabilities for IoT networks. There are a number of existing surveys of IoT middleware.

*Bandyopadhyay et al. (2011b)* and *Bandyopadhyay et al. (2011a)* review a number of middleware systems designed for IoT systems. While they look at security in passing, there is no detailed analysis of the security of each middleware system. *Chaqfeh & Mohamed (2012)* calls out the need for security, but no analysis of the approaches or existing capabilities is provided. *Atzori, Iera & Morabito (2010)* is a very broad survey paper that addresses IoT middleware loosely. *Razzaque et al. (2016)* is another wide-ranging survey of IoT middleware that provides a simple analysis of whether the surveyed systems have any support for security or privacy, but does not address detailed requirements.

It is clear then, that a detailed evaluation of security in IoT middleware is a useful contribution to the literature. We therefore identified a set of middleware systems to study.

### Middleware review methodology

This set was identified through a combination of the existing literature reviews on IoT middleware (*Bandyopadhyay et al., 2011b*; *Chaqfeh & Mohamed, 2012*; *Razzaque et al., 2016*) together with our own search for middleware systems that explicitly target IoT scenarios. Some of the systems that were included in these papers we excluded from our list on the basis that they were not middleware. For example, *Chaqfeh & Mohamed (2012)* lists TinyREST (*Luckenbach et al., 2005*) as a middleware, but in fact we considered this paper to be the definition of a standard protocol and therefore we excluded it.

Our search strategy was to use a search for the terms (''IoT'' OR ''Internet of Things'') AND ''Middleware''. We searched only in the subject terms and restricted the search to academic papers written in English. The search was carried out by the Portsmouth University Discovery system which is a metasearch engine. The list of databases that are searched is available at (*University of Portsmouth Library, 2015*). The search was originally

issued on June 6th, 2015, identifying 152 papers. It was repeated on December 1st, 2016, and 213 papers were identified, showing a significant growth in IoT middleware papers over the intervening period.

We then manually reviewed the abstracts of the 213 papers to identify a list of functioning middleware systems as opposed to papers that describe other aspects of IoT without describing a middleware system. This produced a list of 55 middleware systems.

In our study, we looked for the security and privacy requirements listed in 'Summary of the Review of Security Issues'. We also identified if the middleware had a clearly defined security model and/or security implementation. Out of the 54 middleware systems identified, we found that 35 had no published discussion or architecture for security, or such a minimal description that we were not able to identify any support for the selected security requirements. We label these as *non-secured* systems.

## Non-secured systems

We provide a brief description of each of the non-secure middleware systems:

**ASIP** The Arduino Service Interface Programming model (ASIP) (*Barbon et al., 2016*) is a middleware for Arduino hardware.

**ASPIRE** ASPIRE Project (Advanced Sensors and lightweight Programmable middleware for Innovative Rfid Enterprise applications) (*Prasad, 2008*) is a EU-funded project that created an open, royalty-free middleware for RFID-based applications.

**Autonomic QoS Management** *Banouar et al. (2015)* offers a middleware that autonomically manages Quality of Service (QoS) in IoT scenarios. While this does address some aspects related to security (i.e., accuracy and availability), there is no discussion of how security is handled.

**CASCOM** In *Perera & Vasilakos (2016)* a semantically-driven configuration model is built on top of existing middleware systems such as GSN (*Aberer, Hauswirth & Salehi, 2006*). The authors state their intention of addressing privacy in future work.

**CIRUS** CIRUS (*Pham et al., 2016*) is a cloud-based middleware for ubiquitous analytics of IoT data.

**Cloud-based Car Parking Middleware** In *Ji et al. (2014)* the authors describe an OSGi-based middleware for smart cities enabling IoT-based car parking.

**Context Aware Gateway** *Anand (2015)* provides a reference architecture for using context-awareness in IoT scenarios. The middleware itself does not address security or privacy and the authors plan to address this in further work.

**DAMP** In *Agirre et al. (2016)* there is a middleware—Distributed Applications Management Platform—that can configure systems based on Quality of Service characteristics (QoS). These characteristics can include security, but the system itself does not offer any security model.

**Dioptase** Dioptase *Billet & Issarny (2014)* is a RESTful stream-processing middleware for IoT. Dioptase does address one useful aspect for privacy: intermediate stream processing of data, summarisation and filtering. However, there is no detailed security architecture and description and the security model is left as an item of future work.

**EDBO** *Eleftherakis et al. (2015)* describes the Emergent Distributed Bio-Organization: a biologically-inspired platform for self-organising IoT systems.

**EDSOA** An Event-driven Service-oriented Architecture for the Internet of Things Service Execution (*Lan et al., 2015*) describes an approach that utilizes an event-driven SOA.

**EMMA** The Environmental Monitoring and Management Agent (EMMA) is a proposed middleware based on CoAP (*Duhart, Sauvage & Bertelle, 2015*). It does not offer any security architecture.

**GSN** The GSN framework (*Aberer, Hauswirth & Salehi, 2006*) (Global Sensor Networks) defines a middleware for the Internet of Things that requires little or no programming. The security architecture of the system is not described in any detail: there are diagrams of the container architecture which point to proposed places for access control and integrity checks, but unfortunately there is not sufficient discussion to be able to categorize or evaluate the approach taken.

**Hi-Speed USB middleware** *Augustyn, Maślanka & Hamuda (2016)* offers a middleware based on USB.

**Hitch Hiker** Hitch Hiker 2.0 (*Ramachandran et al., 2015*) is a prototype middleware environment built on Contiki OS.

**LMTS** In *Mhlaba & Masinde (2015)* a middleware system for asset tracking (Laptop Management and Tracking System) is described.

**Middleware for Environmental Monitoring and Control** *Xu, Li & Liang (2016)* defines a middleware for environmental monitoring and control.

**Middleware for Industrial IoT** *Ungurean, Gaitan & Gaitan (2016)* describes a middleware for Industrial IoT based on OpenDDS, which is a middleware that implements the Data Distributions Services (DDS) protocol. At the time of writing the DDS security model was in development and hence the architecture does not address security.

**MIFIM** Middleware for Future Internet Models (MIFIM) (*Balakrishnan & Sangaiah, 2016*) is a Web Service-based architecture that uses Aspect-Orientation to allow for simpler reconfiguration.

**MOSDEN** MOSDEN (Mobile Sensor Data Processing Engine) (*Perera et al., 2014*) is an extension of the GSN approach (see above) which is explicitly targeted at *opportunistic* sensing from restricted devices.

**M-Hub** *Talavera et al. (2015)* describes a middleware for Mobile IoT applications built on top of another middleware (Scalable Data Distribution Layer). In *Gomes et al. (2015)* this work is enhanced to create a middleware for Ambient Assisted Living. In *Vasconcelos et al. (2015)* there is another middleware based on M-Hub. There is no support for security or privacy described.

**PalCom** Palcom (*Svensson Fors et al., 2009*) is a middleware designed for pervasive computing, including IoT systems. It supports ad-hoc composition of services. There is no discussion of security beyond a statement that traditional security models may be added in future.

**POBICOS** Platform for Opportunistic Behaviour in Incompletely Specified, Heterogeneous Object Communities (POBICOS) (*Tziritas et al., 2012*) is a device middleware designed to

run on small devices. In *Tajmajer et al. (2016)* there is a description of migrating aspects of the middleware to a proxy to enable support for smaller devices.

**PROtEUS** PROtEUS is a process manager designed to support Cyber-Physical Systems (*Seiger, Huber & Schlegel, 2016*). It describes a middleware for complex self-healing processes.

**RemoteU¡** *Carvalho & Silva (2015)* offers a middleware for Remote user interfaces.

**SBIOTCM** In *A SOA Based IOT Communication Middleware* (*Zhiliang et al., 2011*) is a middleware based on SOAP and WS. There is no security model described.

**Service Oriented access for Wireless Sensor Networks** *Fronimos et al. (2016)* provides a service-oriented middleware for IoT and Wireless Sensor Network data.

**Smart Object Middleware** *Hernández & Reiff-Marganiec (2015)* describes a Smart Object middleware based on Java.

**symbIoTe** In *Soursos et al. (2016)* a roadmap is laid out for a new EU funded project to allow vertical IoT platforms to interoperate and federate. There is no plan for security presented.

**Thingsonomy** Thingsonomy (*Hasan & Curry, 2015*) is an event-based publish–subscribe based approach that applies semantic technology and semantic matching to the events published within the system.

**UBIROAD** The UBIROAD middleware (*Terziyan, Kaykova & Zhovtobryukh, 2010*) is a specialization of the UBIWARE project specifically targeting traffic, road management, transport management and related use-cases.

**UBISOAP** ubiSOAP (*Caporuscio, Raverdy & Issarny, 2012*) is a Service-Oriented Architecture (SOA) approach that builds a middleware for Ubiquitous Computing and IoT based on the Web Services (WS) standards and the SOAP protocol.

**VEoT** *Alessi et al. (2016)* describes a Virtual Environment of Things which is a middleware for Virtual Reality engagement with the Internet of Things.

**WHEREX** WhereX (*Giusto et al., 2010*) is an event-based middleware for the IoT.

## SECURED SYSTEMS

We identified 19 middleware systems that implement or describe sufficient security architecture that we could evaluate them against the requirements that were identified in 'Summary of the Review of Security Issues'. We describe these systems as *secured*. In addition to the requirements identified above, we also identified whether the systems had explicit support or adaptation for IoT specific protocols: MQTT, CoAP, DDS, Bluetooth or Zigbee. As discussed above, in 'Matrix Evaluation', these protocols have been specifically designed for low-power devices. We label this requirement **REQ7**. Table 2 shows the summary of this analysis.

For each of the secured middleware system we looked at the core published papers and also examined any further available documentation. Below are the specific details of each middleware system.

Fremantle and Scott (2017), *PeerJ Comput. Sci.*, DOI 10.7717/peerj-cs.114

**Table 2  Summary of reviewed middleware systems and major properties.**

| | REQ1—integrity and confidentiality | REQ2—access control | REQ2.1—consent | REQ2.2—Policy-based security | REQ3—authentication | REQ3.1—federated identity | REQ3.2—secure device identity | REQ3.3—anonymous identities | REQ4 -attestation | REQ5—summarisation and filtering | REQ6—context-based security/reputation | REQ7—IoT-specific protocol support |
|---|---|---|---|---|---|---|---|---|---|---|---|---|
| &Cube | Y | Y | | | Y | | | | | | | Y |
| Device Cloud | Y | Y | Y | | Y | Y | | | | | | Y |
| DREMS | Y | Y | | | Y | | | | | | | Y |
| DropLock | | Y | Y | | Y | Y | | | | | | Y |
| FIWARE | Y | Y | Y | Y | Y | Y | | | | | | Y |
| Hydra/Linksmart | Y | Y | | | Y | | Y | | | | | |
| Income | Y | Y | | Y | Y | | | | | | Y | |
| IoT-MP | Y | | | | Y | | | | | | | |
| NERD | Y | | | | Y | | | | | | | Y |
| NOS | Y | Y | | | Y | | | | | | Y | Y |
| OpenIoT | | | | | Y | Y | | | | | | |
| SensorAct | | Y | | Y | | | | | | | | |
| SIRENA | Y | | | | Y | | | | | | | |
| SMEPP | Y | Y | | | Y | | | | | | | |
| SOCRADES | Y | Y | | | Y | | | | | | | |
| UBIWARE | | | | Y | | | | | | | | |
| WEBINOS | Y | Y | | Y | Y | Y | Y | | | | | |
| XMPP | Y | Y | | | Y | Y | | | | | | |
| VIRTUS | Y | Y | | | Y | Y | | | | | | |

### &Cube

In *Yun et al. (2015)* they describe a middleware, &Cube, that is designed to offer RESTful APIs as well as MQTT connections to integrate with IoT devices. The system offers a security manager providing encryption, authentication and access control. No further details are available on the techniques used.

### Device cloud

In *Renner, Kliem & Kao (2014)* there is a blueprint for a middleware that applies Cloud Computing concepts to IoT device middleware. A more detailed exposition is given in *Kliem (2015)*. The approach supports OAuth2.0 to provide tokens to devices. It also supports encryption and access control. There is no support for summarisation, filtering, or consent-based access control described in the publications.

### DREMS

Distributed RealTime Managed Systems (DREMS) (*Levendovszky et al., 2014*) is a combination of software tooling and a middleware runtime for IoT. It includes Linux Operating System extensions as well. DREMS is based on an actor (*Agha, 1985*) model has a well-defined security model that extends to the operating system. The security model includes the concept of multi-level security (MLS) for communications between a device and the actor. The MLS model is based on *labelled* communications. This ensures that data can only flow to systems that have a higher *clearance* than the data being transmitted. This is a very powerful security model for government and military use-cases. However, this approach does not address needs-based access control. For example, someone with *Top Secret* clearance may read data that is categorised as *Secret* even if they have no business reason to utilise that data. The weaknesses of this model have been shown with situations such as the Snowden revelations.

### DropLock

In *Le Vinh et al. (2015)* the authors describe a middleware specifically built for IoT systems and Smart Cities. The DropLock system is designed to enable secure smartphone access to a smart locker, allowing delivery personnel secure access to drop off packages. The system uses secure tokens to allow access to devices. The tokens are passed to the secure locker using Bluetooth.

### FIWARE

FIWARE (*Glikson, 2011*) is a middleware designed to be the basis of a Future Internet, sponsored by the European Union under the FP7 programme. FIWARE is one of the few systems that claim to have used PBD as a basis for design (*Vázquez et al., 2011*). FIWARE has a concept of plugins, known as Generic Enablers (GE). The security model is implemented through GEs including the Identity Management (IdM GE), the Authorization Policy Decision Point (PDP) GE, and the Policy Enforcement Point (PEP) Proxy. The standard approach within FIWARE is based on OAuth2 and XACML. It also supports interoperable standards for exchanging identities with other systems. The overall security design of FIWARE fits into modern authentication and authorization models. IoT devices are

catered for in the FIWARE Architecture through a gateway model. The IoT devices connect to the gateway using IoT specific protocols. The gateway is part of the *IoT Edge*. This communicates via the standard FIWARE protocols into an *IoT Backend* where there are components supporting Device Management and Discovery. The FIWARE documentation does not describe any specific adaptation of security or support for security between devices and the gateway.

### Hydra/Linksmart

Hydra (*Eisenhauer, Rosengren & Antolin, 2009*) was a European Union funded project which has since been extended and renamed as LinkSmart. The Hydra team published a detailed theoretical model of a policy-based security approach (*Adetoye & Badii, 2009*). This model is based on using lattices to define the flow of information through a system. This model provides a language-based approach to security modelling. However, whilst this paper is published as part of the Hydra funded project, there is no clear implementation of this in the context of IoT or description of how this work can benefit the IoT world. Hoever, because Hydra/Linksmart is an Open Source project (*Linksmart, 2015b*) with documentation beyond the scientific papers, it is possible to understand the security model in greater detail by review of this project.

The Hydra and LinkSmart architectures are both based on the Web Services (WS) specifications, building on the SOAP protocol (*Gudgin, 2003*), which in turn builds on the XML Language (*Bray, 2004*). The security model is described in some detail in the LinkSmart documentation (*Linksmart, 2015a*). The model utilises XML Security (*Dournaee, 2002*). There are significant challenges in using this model in the IoT world, as discussed above in 'Matrix Evaluation'. The Hydra/Linksmart approach also uses symmetric keys for security which is a challenge for IoT because each key must be uniquely created, distributed and updated upon expiry into each device creating a major key management issue.

Hydra/Linksmart offers a service called the TrustManager. This is a system that uses the cryptographic capabilities to support a trusted identity for IoT devices. This works with a Public Key Infrastructure (PKI) and certificates to ensure trust. Once again there are challenges in the distribution and management of the certificates to the devices which are not addressed in this middleware.

The Hydra middleware does not offer any policy based access control for IoT data, and does not address the secure storage of data for users, nor offer any user-controlled models of access control to user's data.

In *Patti et al. (2016)* there is a specific instantiation of LinkSmart applied to energy efficiency in buildings. There is no further extension to the security model.

### INCOME

INCOME (*Arcangeli et al., 2012*) is a framework for multi-scale context management for the IoT, funded by the French National Research Agency. The aim of INCOME is to fuse together context data from multiple levels to provide a high-level set of context data from IoT systems that can be applied to decision making, including trust, privacy and security decisions. MuDebs and MuContext (*Lim et al., 2016*) are frameworks built on top of

INCOME that add Attribute Based Access Control (ABAC) and Quality of Context (QoC) processing. MuDebs utilises XACML policies to implement ABAC. MuContext validates QoC and enables privacy filtering.

### IoT-MP

The IoT Management Platform (IoT-MP) is a middleware system described in *Elkhodr, Shahrestani & Cheung (2016)*. IoT-MP offers a security module that implements attribute-based access control (ABAC) against systems. IoT-MP has a model whereby an *Agent* is registered for each class of *Things*, creating the concept of a *Managed Thing*. Agents have unique secure identities. The IoT-MP does not define how devices are identified to agents.

### NAPS

The *Naming, Addressing and Profile Server* (NAPS) (*Liu, Yang & Liu, 2014*) describes a heterogeneous middleware for IoT based on unifying data streams from multiple IoT approaches. Based on RESTful APIs, the NAPS approach includes a key component handling Authentication, Authorization, and Accounting (AAA). The design is based on the Network Security Capability model defined in the ETSI M2M architecture (*ETSI, 2015*). However, the main details of the security architecture have not yet been implemented and have been left for future work. There is no consideration of federated identity or policy based access control.

### NERD

No Effort Rapid Development (NERD) (*Czauski et al., 2016*) is a middleware designed for human IoT interfaces, especially around Bluetooth LE systems and iBeacon discovery. It does not add any new security measures but uses the existing security models in Bluetooth and HTTP.

### NOS

NetwOrked Smart objects (NOS) (*Sicari et al., 2016a*; *Sicari et al., 2016b*) takes an interesting approach to security where the aim is to provide each item of data with a reputational score based on a quality analyser and a security analyser. A machine learning algorithm is used to learn the behaviour of systems in the network and adjust the scores based on the potential attacks and the applied countermeasures. The system incorporates keys and key-based authentication, encryption and complex passwords.

### OpenIoT

OpenIoT is an open cloud-based middleware for the Internet of Things, funded by the European Union FP7 programme. It also extends the GSN framework. The *Security Module* uses OAuth2 as the main authentication and authorization model for web-based systems. No details are given of how sensors are authenticated or authorized.

### SensorAct

SensorAct (*Arjunan et al., 2015*) is an IoT middleware specifically aimed at providing support for Building Management Systems (BMS). It supports fine-grained access control through the use of a rules engine to implement access control policies. No details are provided of the authentication models at the device level or the web interface.

## SIRENA

SIRENA (Service Infrastructure for Real-time Embedded Networked Devices) (*Bohn, Bobek & Golatowski, 2006*) is a SOAP/WS-based middleware for IoT and embedded devices. While there is little description of the security framework in SIRENA, it does show the use of the WS-Security specification. As previously discussed, this approach is very heavyweight, has issues with key distribution, federated identity and access control.

## SMEPP

Secure Middleware for P2P (SMEPP) (*Benito et al., 2009*) is an IoT middleware explicitly designed to be secure, especially dealing with challenges in the peer-to-peer model. SMEPP security is based around the concept of a group. When a peer attempts to join a group, the system relies on challenge-response security to implement mutual authentication. At this point the newly joined peer is issues a shared session key which is shared by all members of the group. SMEPP utilizes elliptic key cryptography to reduce the burden of the security encryption onto smaller devices. Overall SMEPP has addressed security effectively for peer-to-peer groups, but assumes a wider PKI infrastructure for managing the key model used within each group. In addition, there is no discussion of access control or federated identity models, which are important for IoT scenarios. The model is that any member of the group can read data published to the group using the shared session key.

## SOCRADES

SOCRADES (*De Souza et al., 2008*) is a middleware specifically designed for manufacturing shop floors and other industrial environments. Based on SOAP and the WS stack it utilizes the security models of the WS stack, in particular the WS-Security standard for encryption and message integrity. There is no special support for federation, tokens or policy-based access control (instead relying on role-based access control). The resulting XML approach is very heavyweight for IoT devices and costly in terms of network and power (*Dunkels & Yazar, 2009*). In addition, the lack of explicit support for tokens and federated security and identity models creates a significant challenge in key distributions and centralized identity for this approach.

## UBIWARE

The UBIWARE project is a smart semantic middleware for Ubiquitous Computing (*Scuturici et al., 2012*). The security model for UBIWARE is not clearly described in the original paper, but an additional paper describes a model called Smart Ubiquitous Resource Privacy and Security (SURPAS) (*Naumenko, Katasonov & Terziyan, 2007*), which provides a security model for UBIWARE. UBIWARE is designed to utilize the semantic Web constructs, and SURPAS utilises the same model of semantic Web as the basis for the abstract and concrete security architectures that it proposes. The model is highly driven by policies and these can be stored and managed by external parties. In particular the SURPAS architecture is highly dynamic, allowing devices to take on board new roles or functions at runtime. While the SURPAS model describes a theoretical solution to the approach, there are few details on the concrete instantiation. For example, while the model defines a policy-based approach to access control, there are no clearly defined policy languages

chosen. There is no clear model of identity or federation, and there is no clear guidance on how to ensure that federated policies that are stored on external servers are protected and maintain integrity. The model does not address any edge computing approaches or filtering/summarisation of IoT data. However, the overall approach of using ontologies and basing policies on those ontologies is very powerful.

## WEBINOS

The Webinos (*Desruelle et al., 2012*) system has a well-thought through security architecture. The documentation explicitly discussed PBD. The Webinos system is based around the core concept of devices being in the personal control of users and therefore having each user having a personal zone to protect. This is a more advanced concept but in the same vein as the protected sub-domains in VIRTUS. In the Webinos model, each user has a cloud instance—known as the Personal Zone Hub (PZH) that supports their devices. The Personal Zone Hub acts as a service to collect and offer access to data and capabilities of the user's devices. The PZH acts as a certificate authority, issuing certificates to the devices that are used for mutual authentication using TLS. User's authenticate to the PZH using the OpenID protocol. On the device, a communications module known as the Personal Zone Proxy (PZP) handles all communications with the PZH.

The idea of the Personal Zone may have significant issues however, when a single device is used by many different people (for example, the in-car system in a taxi as opposed to a personal vehicle). These issues are not addressed in Webinos, though they are called out in the lessons learnt.

Webinos utilizes policy-based access control modelled in the XACML (*Godik et al., 2002*) language. The system pushes XACML policies out to devices to limit the spread of personal and contextual data.

Webinos addresses the issue of software modification using an attestation API, which can report whether the software running is the correct level. This requires the device to be utilising Trusted Platform Module (TPM) hardware that can return attestation data.

Webinos also addresses the issue of using secure storage on devices where the device has such storage.

While the Webinos project does address many of the privacy concerns of users through the use of the Personal Zone Hub, there is clearly further work that could be done. In particular the ability for users to define what data they share with other users or other systems using a protocol such as OAuth2 (*Hammer-Lahav & Hardt, 2011*), and the ability to install filters or other anonymising or data reduction aggregators into the PZH are lacking. One other aspect of Webinos that is worth drawing attention to is the reliance on a certain size of device: the PZP that is needed on the device is based on the *node.js* framework and therefore the device needs to be of a certain size (e.g., a 32-bit processor running a Linux derivative or similar) to participate in Webinos.

## VIRTUS

The VIRTUS middleware (*Conzon et al., 2012*) utilizes the core security features of the XMPP protocol to ensure security. This includes tunnelling communications over TLS,

authentication via SASL, and access control via XMPP's built-in mechanisms. SASL is a flexible mechanism for authentication which supports a number of different systems including token-based approaches such as OAuth2 or Kerberos, username/password, or X.509 certificates. For client-to-server based communications, it is not clear from the description which of these methods is actually implemented within VIRTUS. For server-to-server communications there is specified the use of SASL to ensure full server federation.

While the VIRTUS model does not describe the challenges of implementing a personal instance of middleware for single users or devices, there is a concept of edge computing described, where some interactions may happen within an edge domain (e.g., within a house) and lower security is required within that domain while higher security is expected when sharing that data outside. This model is fairly briefly described but provides an interesting approach. One challenge is that there are multiple assumptions to this: firstly, that security within the limited domain needs less security, when there may be attackers within that perimeter. Secondly, that the open channel to the wider Internet cannot be misused to attack the edge network. The ability to calculate, summarise and/or filter data from the edge network before sharing it is also not discussed except in very granular terms (e.g., some data are available, other data are not).

### XMPP

The paper (*Iivari et al., 2014*) describes how the XMPP architecture can be applied to the challenges of M2M and hence the IoT, together with a proof-of-concept approach. The system relies on the set of XMPP extensions around publish/subscribe and the related XMPP security models to implement security. This includes TLS for encryption, and access control models around publish–subscribe. There is also a discussion about leakage of information such as *presence* from devices. The proof-of-concept model did not include any federated identity models, but did utilize a One-Time Password (OTP) model on top of XMPP to address the concepts such as temporary loans of devices.

## SUMMARY OF IOT MIDDLEWARE SECURITY

In reviewing both the security and privacy challenges of the wider IoT and a structured review of more than fifty middleware platforms, we have identified some key categories that can be applied across these areas.

Firstly, we identified the significant proportion of the systems that did not address security, left it for further work, or did not describe the security approach in any meaningful detail. There were other systems (such as UBIWARE and NAPS) that offer theoretical models but did not demonstrate any real-world implementation or concrete approach.

The next clear category are those middlewares that apply the SOAP/Web Services model of security. This includes SOCRADES, SIRENA, and Hydra/Linksmart. As we have discussed in the previous sections there are significant challenges in performance, memory footprint, processor power and usability of these approaches when used with the IoT.

Two of the approaches delegate the model to the XMPP standards: VIRTUS and XMPP (*Conzon et al., 2012*; *Iivari et al., 2014*). XMPP also has the complexity of XML, but avoids the major performance overheads by using TLS instead of XML Encryption and XML

Security. In addition, recent work on XMPP using EXI makes this approach more effective for IoT.

This finally leaves a few unique approaches, each of which brings their own unique benefits.

DREMS is the only system to provide Multi-level security based on the concept of security clearances. While this model is attractive to government and military circles (because of the classification systems used in those circles), we would argue that it fails in many regards for IoT. In particular there are no personal controls, no concept of federated identity and no policy based access controls in this model.

SMEPP offers a model based on public key infrastructures and shared session keys. We would argue this approach has a number of challenges scaling to the requirements of the IoT. Firstly, there are significant issues in key distribution and key revocation. Secondly, this model creates a new form of perimeter—based on the concept of a shared session key. That means that if one device is compromised then the data and control of all the devices in that group are also compromised.

Only Dioptase supports the concept of stream processing in the cloud, which we argue is a serious requirement for the IoT. The requirement is to be able to filter, summarise and process streams of data from devices to support anonymisation and reduction of data leakage.

FIWARE has a powerful and extensible model for authentication and access control, including support for federated identity and policy-based access control.

Finally, the we identified that the most advanced approach is that proposed by Webinos. Webinos utilizes some key technologies to provide a security and privacy model. Firstly, this uses policy-based access control (XACML). The model does not however support user-guided access control mechanisms such as OAuth2 or UMA.

Webinos does support the use of Federated Identity tokens (OpenID), but only from users to the cloud, as opposed to devices to the cloud. We and others have proposed the model of using federated identity tokens from the device to the cloud in *Fremantle et al. (2014)*, *Fremantle, Kopecký & Aziz (2015)* and *Cirani et al. (2015)*.

The contribution of the Webinos work with the largest potential impact is the concept of Personal Zone Hub, which is a cloud service dedicated to a single user to handle the security and privacy requirements of that user. There is, however, further research around this area: the PZH model from Webinos does not examine many of the challenges of how to implement the PZH in real life. For example, user registration, cloud hosting, and many other aspects need to be defined in more detail before the Webinos PZH model is practicable for real world projects. In addition there are challenges using the PZH model with smaller devices, because of the requirement to use the PZP.

## Overall gaps in the middleware

When we look at the requirements for security and privacy of the Internet of Things we can see there are some gaps that are not provided by any of the reviewed middleware systems.

- Only two of the middleware systems explicitly applied the concept of PBD in designing a middleware directly to support privacy, although Webinos did exhibit many of the characteristics of a system that used this approach.
- Only two of the systems applied any concepts of context-based security or reputation to IoT devices.
- User consent was only supported in three of the systems.
- None of the systems supported anonymous identities or attestation.
- None of the systems satisfied all the requirements identified.

## DISCUSSION

### Contributions

In this paper we have taken a two-phase approach to reviewing the available literature around the security and privacy of IoT devices.

In the first part we created a matrix of security challenges that applied the existing CIA+ model to three distinct areas: device, network and cloud. This new model forms a clear contribution to the literature. In each of the cells of the matrix we identified threats, challenges and/or approaches, or in a few cells we identified that the challenges are not exacerbated by IoT concerns. We further used Spiekerman and Cranor's three layer privacy model to analyse the privacy requirements of IoT. We used this analysis to identify seven major requirements and five subsidiary requirements.

In the second part, we used a structured search approach to identity 54 specific IoT middleware frameworks and we analysed the security models of each of those. We utilised the twelve requirements from the first phase to validate the capabilities of each system. While there are existing surveys of IoT middleware, none of them focussed on a detailed analysis of the security of the surveyed systems and therefore this has a clear contribution to the literature.

### Further work

In our survey, we have identified some clear gaps. Over half the surveyed systems had either no security or no substantive discussion of security. Out of the surveyed systems we found very few that addressed a significant proportion of the major challenges that we identified in the first section. We found certain aspects that were identified in the first section that were not addressed by any of the surveyed systems. Based on this we believe there is a significant opportunity to contribute to the research by creating a middleware for IoT that addresses these gaps.

- To define a model and architecture for IoT middleware that is designed from the start to enable privacy and security (Privacy by Design).
- Secondly, to bring together the best practice into a single middleware that includes: federated identity (for users and devices), policy-based access control, user managed access to data, stream processing in the cloud.
- Thirdly, there is considerable work to be done to define a better model around the implementation challenges for the concept of a personal cloud service (e.g the Webinos

PZH). This includes the hosting model, bootstrapping, discovery and usage for smaller devices.

- Finally, creating a middleware system that applies context-based security and reputation to IoT middleware.

### Funding
The authors received no funding for this work.

### Competing Interests
The authors declare there are no competing interests.

### Author Contributions
- Paul Fremantle conceived and designed the experiments, performed the experiments, analyzed the data, wrote the paper, prepared figures and/or tables, performed the computation work, reviewed drafts of the paper.
- Philip Scott reviewed drafts of the paper.

### Data Availability
The raw data is included in the tables in the manuscript.

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
