# Peer review of "A survey of secure middleware for the Internet of Things"

_PeerJ Computer Science, doi:10.7717/peerj-cs.114_

## Round 0.1 · original submission · Major Revisions

Based on the reviewer's comments, this article should go through a major revision before it can be published.

Reviewer 1 ·

Basic reporting

The introduction is not strong. It only has some basic information about IoT, but nothing related to the security of the IoT and how the work contributes to the field.

In addition, the section 4 (summary and conclusion) doesn't have actual conclusion of the survey.

Experimental design

There is no need to discuss a list of the middle-ware with insufficient description of the security. It would be enough to just summarize how many middle-wares reviewed did not discuss the security.

In addition, this paper has two parts that are kind of disconnected. It first defines the security matrix, and then discusses the security design/implementation of reviewed middle-wares. However, the security design/implementation discussion has little emphasis on the challenges in the security matrix.

Validity of the findings

The findings are not well summarized.

Additional comments

I would suggest removing those middle-wares without any security information. On the other hand, it will be useful to have more insightful discussions for those with the security design and implementation. For example, there are several middle-ware using XML that is argued to have substantial performance overhead. However, the XML is already used by those middle-wares. Is there any performance study in original middle-ware design to support the use of XML? Do those middle-wares have poor performance because of XML? Do they have more than enough hardware power?

Reviewer 2 ·

Basic reporting

This paper reviews the literature on the challenges and approaches to security and privacy in the internet of things, with an especial focus on how these aspects are handled in IoT middleware. The topic is important and it is an interesting paper.

Experimental design

This paper mainly reviews the status of IoT middleware. It does not have design with high novelty and the related experiments.

Validity of the findings

This paper presents the states and the challenges of IoT middleware from different perspectives. It does not have a lot of data or result. The conclusion is solid and appropriately stated.

Additional comments

This paper reviews the the current literature on the challenges and approaches to security and privacy in the Internet of Things. The content is solid and the conclusion is stated appropriately .

---

## Round 0.2 · accepted · Accept

The revision has addressed the concerns from the previous reviewers, and thus I recommend the manuscript to be accepted.